# Infrared thermal imaging monitoring on hands when performing repetitive tasks: An experimental study

Alejandra García Becerra[1], Jesús Everardo Olguín-Tiznado[2], Jorge Luis García Alcaraz[3]*, Claudia Camargo Wilson[2], Blanca Rosa García-Rivera[4], Ricardo Vardasca[5,6,7], Juan Andres López-Barreras[8]

1 Industrial Engineering Department, Tecnológico Nacional de México/I.T Cd. Guzman, Jalisco, México, 2 Faculty of Engineering, Arquitecture and Design, Autonomous University of Baja California, Ensenada, Mexico, 3 Department Industrial Engineering and Manufacturing, Autonomous University of Ciudad Juarez, Ciudad Juárez, Mexico, 4 Faculty of Administrative and Social Sciences, Autonomous University of Baja California, Valle Dorado, México, 5 Faculdade de Engenharia, Universidade do Porto, Porto, Portugal, 6 INEGI, Universidade do Porto, Porto, Portugal, 7 ISLA Santarém, Santarém, Portugal, 8 Faculty of Chemical Sciences and Engineering, Autonomous University of Baja California, Tijuana, Mexico

* jorge.garcia@uacj.mx

**Data Availability Statement:** The data underlying this study are available on Figshare (https://doi.org/10.6084/m9.figshare.14057966).

## Abstract

The monitoring of infrared thermal images is reported to analyze changes in skin temperature in the hand fingers when repetitive work is performed to know which finger has a greater risk of injury, besides, the recovery time is analyzed regarding the initial temperature and its relationship with age, sex, weight, height if practice sports, and Body Mass Index (BMI) per individual. For the above, an experimental test was carried out for 10 minutes on a repetitive operation that takes place in the telecommunications industry and 39 subjects participated in which an infrared thermal image of the dorsal and palmar part of both hands was taken in periods of 5 minutes after the 10-minute test has elapsed. The results show that none of the participants recovered their initial temperature after 10 minutes of the experimental test. In addition, it was found that there is a relationship between skin temperature and sex, and that age influences the recovery of temperature. On the other hand, the thumb, index, and middle fingers have a higher risk of injury in the analyzed task. It is concluded that performing repetitive work with all the fingers of the hand does not show that all they have the same risk of injury, besides that, not all the variables studied affect the recovery of temperature and its behavior.

## 1. Introduction

Musculoskeletal Disorders (MSD) are the consequence of highly repetitive tasks [1, 2]. The problems caused by MSDs are of growing interest and represent painful, debilitating, and long-lasting effects in many cases [3]. The importance of studying MSDs is that there were approximately 2.8 million non-fatal injuries and illnesses in the workplace reported by private industry employers in 2017, according to the United States Bureau of Labor Statistics.

**Funding:** The authors did not receive any specific funding for this work.

**Competing interests:** The authors have declared that no competing interests exist.

**Abbreviations: BMI**, Body Mass Index; **DUE**, Disorders of the Upper Extremities; **IT**, Infrared Thermography; **LR**, left wrist; **MSD**, Musculoskeletal Disorders; **ROI**, Regions of Interest; **RSI**, Repetitive Strain Injury; **RW**, right wrist; **SCI or MSD**, Musculoskeletal Injuries or Disorders; **SOII**, Survey of Occupational Injuries and Illnesses.

However, they reported 45,800 fewer cases of non-fatal injuries and illnesses compared to the previous year, according to estimates from the Survey of Occupational Injuries and Illnesses (SOII) [4]. Likewise, work-related injuries and illnesses cause the loss of 3.9% of all years of work in the world and 3.3% of the European Union (EU), which is equivalent to a cost of approximately 2,680 billion and 476 billion dollars respectively [5]. One of the consequences of repetitive work is MSDs, which represent one of the most important problems in the European Union and Latin America [6].

Repetitive upper extremity work is considered one of several physical workload factors associated with symptoms and injuries of the musculoskeletal system. The development of a MSD is related to different factors of exposure at work, such as vibration, excessive forces, inappropriate postures, and repetitive movements [7]. Therefore, it is concluded that the problems caused by repetitive activities are increasingly worrying and have painful consequences for the worker [3].

MSD problems have economic repercussions for workers, the company, and health institutions [6, 8]. The effects of diseases due to exposure in the workplace affect people's quality of life, the loss of working hours and the professional reconversion of new employees, indirect costs that include high absenteeism, high turnover, low employee morale [9] and they correlated with the high costs generated by medical care and days of disability [1, 10].

One of the techniques that help to identify MSD is Infrared Thermography (IT), which is a non-invasive technique without biological risk, which detects, measures, and converts the invisible surface body heat into a visible screen, which is then photographed or videotaped as a permanent record. Also, it graphically represents the temperature distribution on a given body surface at a given time and has been used to study biological thermoregulatory abnormalities that directly or indirectly influence skin temperature [11, 12]. On the other hand, an infrared thermogram is an image of the temperature distribution of the target to be analyzed [13]. In summary, thermography captures the natural thermal radiation generated by an object at a temperature above absolute zero [14] and is used in the medical field to provide information on the physiological responses associated with skin temperature and to identify different types of pain syndromes [15, 16].

MSD is a term that groups together diseases that affect different parts of the locomotor system caused by its excessive use [17], caused by prolonged and tiring work and postures, improper unbalanced angles, load handling and repetitive movements [18]. DUE are accumulated lesions that frequently occur in connective soft tissues, particularly to tendons and their sheaths. An MSD can irritate or damage nerves and impede blood flow through arteries and veins. They are recurrent in the wrist area, shoulder, and neck. For example, Carpal Tunnel Syndrome (CTS), tendinitis, tendosynovitis, Guyon's tunnel syndrome, are diseases resulting from repetitive efforts within the work occupation [19].

Since 1960, infrared thermal imaging has contributed to diagnostics in medicine [20]. Currently, several studies to diagnose Musculoskeletal Injuries or Disorders (SCI or MSD) through the application of this technique and as in Symons, Byiers [12], since the body temperature is an indicator that any pathological problem is present [20] and as Ring [21] indicates, thermography is effective in studying the distribution of skin temperature, as "the physiological mechanisms of temperature distribution on the surface of the body are now better understood" and precise tests based on thermal images of some definite disorders [13]. Human skin has an emissivity of 0.98 and Charlton, Stanley [22] tested 65 people to see if skin color affected thermography precision results, and found that this was not significant, in other words, skin pigmentation does not affect the thermal emissivity measurement.

It is now possible to generate a preliminary diagnosis from thermal images since abnormal thermal patterns are easily identifiable by IT [23]. Another advantage is that the IT can

measures the temperature distribution through an image and monitor the surface of interest [24]. Gold et al. [25] showed that there is a moderate correlation between blood flow volume and mean skin temperature during a typing task in which the speed of activity affects both factors. Besides, skin temperature is affected by blood perfusion; therefore, diseases that affect blood circulation, especially in the extremities, can be studied with this technique [26]. The hypothalamus controls body temperature and determines the value to which it will regulate [27], it balances heat generation with heat loss, is connected to the pituitary gland at the base of the brain near the termination of the brainstem [14].

Sharma, Smith [28] have applied a 5-minute typing stress test in 21 patients with Repetitive Strain Injury (RSI). All 21 patients were in pain after 5 minutes of typing. The post-test mean temperature reading was significantly reduced by 2.11 ˚C with a 95% confidence interval of 1.35 to 2.26 ˚C (P<0,001).

Madeleine, Voigt [29] have developed a laboratory study to examine subjective, physiological, and biomechanical responses to repetitive manual light work in prolonged time while staying on soft, and hard surfaces. The results highlighted a greater feeling of comfort when standing on the soft surface. Also, postural activity was lower when standing on the soft surface, but the activity was sufficient to prevent swelling of the lower legs. Looked at repetitive hand movements, where the results explained that participants without task-related pain showed less variability between cycles of a task compared to healthy controls.

Other studies have focused on prolonged manual work, for example, Camargo, Ordorica [11] have applied three hours and thirty minutes of an experimental test of a repetitive operation of the textile industry for three days in two healthy right-handed people, with a controlled temperature in the laboratory between 20 and 25 ˚C, with 20 minutes of stabilization time in the beginning and at the end of the operation. The results showed that the maximum temperature obtained in the right wrist (RW) was 35.078 ˚C during a period of 1 hour 41 minutes 52 seconds; and in the left wrist (LR), 34.663 ˚C during a period of 2 hours 42 minutes 51 seconds, detecting discomfort in the right shoulder and wrist in the time range where the highest temperatures were identified.

However, the information that must be considered in the evaluation and interpretation of thermal images when working with IT must be focused on specific factors [30], while Sharma, Smith [28], Suominen and Asko-Seljavaara [31] and YK Ng [32] recommend specifically analyzing certain factors on the influence of skin temperature.

Ramos, dos Reis [33] in a poultry slaughterhouse, analyzed repetitive work in a cold environment, being the dominant hand the one that registered the highest temperatures, and it was found that combining repetitive work with high temperatures leads to a decrease in body temperature, which results in DUE.

It is important to mention that the studies that have been found regarding repetitive movements with the hands generally refer to writing experiments, where the muscles and joints of the neck, shoulders, arms, and hands are used when working excessively, resulting in cumulative trauma disorders [34], where the parts of the body are studied in their entirety. The contribution of this work is that it analyzes the temperature in the fingers of the hands during repetitive tasks, through infrared thermography, in order to detect which possible SCI may occur, as previous research considers the entire region of the hand without thoroughly determining each area of the fingers. In other words, this research investigates the temperature of each specific finger and determines which of these maintain maximum temperatures for a longer time. This information helps to know which of these has a greater tendency to suffer a MSD.

The manuscript is organized as follows, initially it presents a general introduction to MSD and IT. The materials, methods, and statistical analysis used are shown below. Then the results and discussion are presented, and finally the conclusion and future research on the subject.

## 2. Methodology

### 2.1 Materials

The materials and equipment with which this study was developed were: A personal computer with an Intel Core I3, 4gb Ddr4 processor: i3-6006U, 4 GB of RAM memory. An infrared thermography camera Thermal brand FLIR® (Wilsonville, OR, USA) E25 (Long wave IR camera at 7.5 to 13 μm spectral range, Focal Plane Array sensor of 160x120, Noise Equivalent Temperature Difference of <100mK at 30°C and measurement uncertainty of ±2% of overall reading) was used.

The images were analyzed with FLIR® software (Wilsonville, OR, USA) ThermaCAM Researcher Pro2.10 that allows drawing Regions of Interest (ROI) and extracting average temperatures from them. The ROIs used were the inner canthus of the eyes (to estimate body temperature [35], hands, and fingers).

Data was organized using the Microsoft® (Redmond, WA, USA) Excel 10 spreadsheet. The statistical analysis was done using the IBM SPSS® program (Armonk, NY, USA) statistic software v23, and Minitab 18® program (State College, PA, USA). Currently, data is available in a repository and can be consulted at Garcia Becarra, Olguín Tiznado [36] in this link: https://doi.org/10.6084/m9.figshare.14057966.

### 2.2 Characterization of the sample

The test was carried out with 39 participants (n = 18 women and n = 21 men) with an average age of 22.4 ± 4.7 years, an average height of 1.69 ± 0.08 m, an average weight of 69.24 ± 14.57 kg, and an average Body Mass Index (BMI) of 23.9 ± 3.8 kg/m². The characterization of the sample is presented in Table 1, which contains sex, age, BMI, dominant hand, history of fractures and sports habits.

The purpose of the test and the importance of their participation in the research are explained to the participants. The data collection process followed the ethical principles of the Declaration of Helsinki for Medical Research with Human Subjects, adopted by the 18th General Assembly of the World Medical Association in 2001.

### 2.3 Methods

Every participant in this research was informed previously using a verbal consent at first instance and invited to participate and were recruited in the months of October and November 2017 in a laboratory of the Autonomous University of Baja California. Once they accept to participate, the first day of the experiment, they were again verbally informed regarding the type of research and everyone had signed a consent, authorizing to use data obtained for academic and scientific purposes. The research protocol and the ethic statement were reviewed and approved by the Postgraduate Department Committee of the Faculty of Design and Engineering of Autonomous University of Baja California, according to the Official Mexican Standards NOM-035-STPS-2018, NOM-030-STPS-2009 and NOM-036-1-2018 regarding ergonomic risk in jobs.

The tests were performed in a laboratory with controlled conditions for two months. The exclusion criteria for participants were that they should not have consumed caffeine, drank alcohol, or exercised prior to the test. Before the experimental tests, participants were asked not to drink alcohol, tea, coffee, or smoke or use any ointment on their skin. Furthermore, another requirement for the experiment was that participants did not perform any physical activity during the 20 minutes prior to the start of the test [37, 38], according to the guidelines of thermography to reinforce the heat balance [32]. The participants waited 20 minutes in a

**Table 1. Characterization of the sample.**

| | Sex | Age | Height | Fractures | Practice sports | Weight | BMI |
|---|---|---|---|---|---|---|---|
| 1 | W | 20 | 1.60 | 0 | 0 | 53 | 20.7 |
| 2 | W | 20 | 1.64 | 0 | 0 | 65 | 24.17 |
| 3 | W | 20 | 1.68 | 0 | 1 | 66 | 23.38 |
| 4 | W | 20 | 1.60 | 0 | 0 | 63 | 24.61 |
| 5 | W | 20 | 1.57 | 0 | 0 | 54 | 21.91 |
| 6 | W | 20 | 1.74 | 0 | 1 | 75 | 24.77 |
| 7 | W | 32 | 1.66 | 0 | 0 | 90 | 32.66 |
| 8 | W | 34 | 1.63 | 0 | 0 | 64 | 24.09 |
| 9 | W | 22 | 1.67 | 0 | 1 | 65 | 23.31 |
| 10 | W | 21 | 1.63 | 0 | 0 | 50 | 18.82 |
| 11 | W | 20 | 1.65 | 0 | 0 | 56 | 20.57 |
| 12 | W | 23 | 1.50 | 0 | 0 | 45 | 20 |
| 13 | W | 21 | 1.64 | 0 | 0 | 62 | 23.05 |
| 14 | W | 22 | 1.79 | 0 | 1 | 76 | 23.72 |
| 15 | W | 23 | 1.55 | 0 | 0 | 53 | 22.06 |
| 16 | W | 19 | 1.61 | 0 | 0 | 55 | 21.22 |
| 17 | W | 38 | 1.63 | 0 | 0 | 93 | 35 |
| 18 | W | 21 | 1.67 | 0 | 0 | 72 | 25.82 |
| 19 | M | 20 | 1.73 | 0 | 1 | 70 | 23.39 |
| 20 | M | 20 | 1.75 | 0 | 1 | 70 | 22.86 |
| 21 | M | 21 | 1.74 | 0 | 1 | 63 | 20.81 |
| 22 | M | 20 | 1.68 | 0 | 1 | 63.5 | 22.5 |
| 23 | M | 20 | 1.66 | 0 | 1 | 69 | 25.04 |
| 24 | M | 22 | 1.63 | 0 | 1 | 43 | 16.18 |
| 25 | M | 21 | 1.79 | 0 | 0 | 66 | 20.6 |
| 26 | M | 22 | 1.84 | 0 | 1 | 115 | 33.97 |
| 27 | M | 25 | 1.75 | 0 | 1 | 83 | 27.1 |
| 28 | M | 20 | 1.80 | 1 | 1 | 100 | 30.86 |
| 29 | M | 20 | 1.74 | 0 | 0 | 75 | 24.77 |
| 30 | M | 21 | 1.71 | 0 | 1 | 77 | 26.33 |
| 31 | M | 20 | 1.83 | 0 | 1 | 80 | 23.89 |
| 32 | M | 21 | 1.62 | 0 | 1 | 61 | 23.24 |
| 33 | M | 21 | 1.66 | 0 | 0 | 66 | 23.95 |
| 34 | M | 23 | 1.83 | 0 | 0 | 70 | 20.9 |
| 35 | M | 20 | 1.75 | 0 | 1 | 68 | 22.2 |
| 36 | M | 21 | 1.77 | 0 | 1 | 81 | 25.85 |
| 37 | M | 38 | 1.75 | 0 | 1 | 83.9 | 27.4 |
| 38 | M | 23 | 1.75 | 0 | 0 | 63 | 20.57 |
| 39 | M | 21 | 1.84 | 0 | 0 | 76 | 22.45 |
| **Mean** | | 22.46±4.70 | 1.69±0.08 | | | 69.24±14.57 | 23.96±3.89 |

W–Woman, M–Man

closed room with moderate lighting and without ventilation, at an average temperature, and a constant relative humidity before performing the experimentation tests [39, 40].

The recruitment of the participants took place at the UABC with undergraduate and graduate students and some university workers. The sample is considered representative of a larger

population since it includes both genders between an age range of 19 to 38 years, including a pregnant woman.

The temperature at the room where the tests are applied and the humidity level were verified to prove that the room temperature was maintained between 20 to 23 ˚C and the relative humidity did not exceed 50% [40, 41]. The emissivity value of the infrared camera was fixed at 98%, which is the recommended value for human skin [30].

For each two-hour session, 3 participants were considered to obtain the thermographic images. Each one is explained what the process consists of. Before starting, they were asked if they smoked, were taking any medication, played sports, or had a fracture and the dominant hand, in this case all participants are right-handed. Besides, data on their weight, height, dominant hand were taken and BMI was estimated.

Emulated tests of a work task were carried out in a cable assembly area in a harness assembly company for the automotive industry, the activity was emulated in a laboratory under conditions in which it would work in real life.

The process consisted of rolling a pair of wires by hand for 10 minutes. This movement involves the 10 fingers of the hands and twists of the wrists as shown in Fig 1, the process consisted of rolling a pair of wires by hand for 10 minutes. This movement involves the 10 fingers of the hands and twists of the wrists; however, not all fingers work in the same way. Subsequently, a photo of their face was taken to measure the temperature of the inner canthus of their eyes to make an estimate of the central body temperature, and of the dorsal and palms of the hands before beginning with the repetitive movement Fig 2 shows examples of face, palm of the hand images with the respective ROIs studied and identified and the ROIs considered in the face were the carotids, which are found in the tear duct of the eye, they were delimited and for the hands, each of the regions of the fingers were delimited. The average number pixel in ROIs were 15x43 by thumb, 14x65 by index, 14x66 by middle,14x55 by ring, 14 x47 by little, and 2x3 in tear ducts.

Minute 0 is considered before the start of the repetitive movement and after 10 minutes of activity with repetitive movement, 3 pictures of thermographic images were taken at intervals

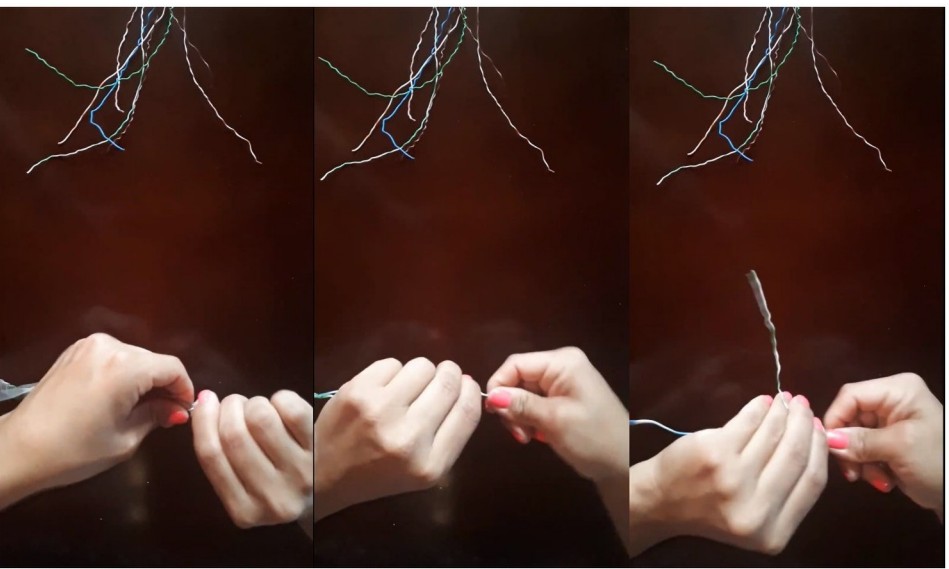

**Fig 1. Steps of activity that participants have carried out.** The process consisted of winding cables without stopping for 10 minutes.

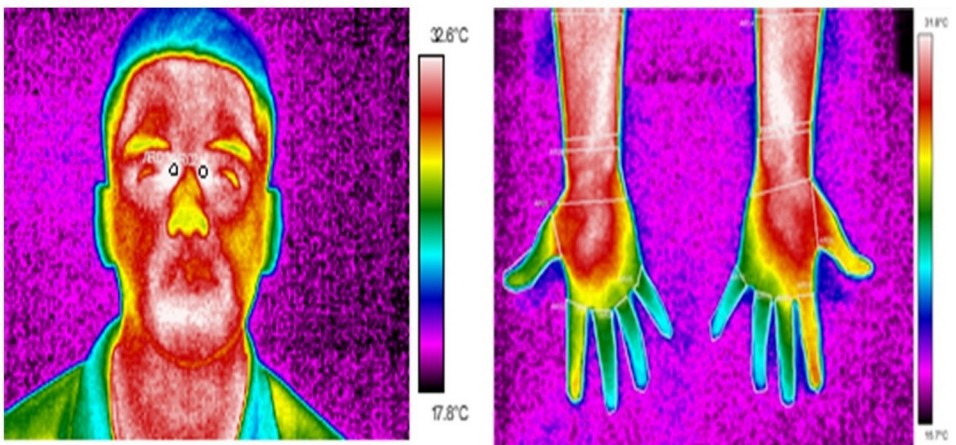

**Fig 2. Thermogram of face and palm of the hand with their respective region of interest.** The ROIs for face were the carotids, which are found in the tear duct of the eye and ROIs for hands were each region delimited by fingers. Resolution average was 160 x 120 pixels.

of 10, 15, and 20 minutes. The hands are placed on a template, maintaining the same posture in each shot.

## 3. Results

### 3.1 Statistical analysis

**3.1.1 Normality test.** The ROIs variables are declared with the nomenclature as follows: *D (dorsal), P (palm area), F (finger), (1 thumb, 2 index, 3 middle, 4 ring and 5 little finger), R (right), L(left)* the end represents time. For example, DF1R_10 refers to the dorsal part of the thumb, of the right hand at 10 minutes, PF1L_15 refers to the palm part of the thumb, of the left hand at 15 minutes. Please see S1 Annexure for complete nomplenclature.

The normality data test for data captured was performed using Anderson Darling test, where not all the variables showed a normal distribution. Variables that are not normal are: DF1R_10, DF1L_10, DF2L_10, DWL_10, DF1R_15 and DF2R_15 for dorsal area and for palmar area: PF1R_10-PF3R_10, PF1L_10, PF2L_10, PWL_10, PFL_10, PF1R_15, PF2R_15, PF1L_15, PF2L_15, PF1R_20-PF4R_20, PF1L_20-PF3L_20. Please see S2 Annexure view all variables and the p-value aasociated.

**3.1.2 Mann-Whitney U test: Variables and ROIs.** To test the effect of variables on ROIs, the Mann Whitney U test is used with a level of significance of 0.05. The variables are sex, age, fractures, paticipants that practice some sports and the temperatures of the ROIs in eyes EL (left eye) and ER (right eye), and the average temperature of both eyes. From results obtained, it is corroborated that sex and temperature of the eyes (right carotid, left carotid and the average of both temperatures) have an influence with some ROIs. Table 2 shows the variables where the alternative hypothesis is accepted. However, it is important to mention that although it is not reported, the variables age, BMI, fractures, and sports does not have an influence on ROIs temperature.

**3.1.3 Mann-Whitney U test between ROIs and the sex variable.** For the analysis of sex in each of the ROIs, it is done with the Mann-Whitney U test, where the results show asymptotic significance as is illustrated in Fig 3, where x-axis indicates the ROIs variable name and y-axis indicates the p-values for a two-tailed hypotheses test. The ROIs variables are declared with the nomenclature as follows: *D (dorsal), P (palm area), F (finger), (1 thumb, 2 index, 3*

**Table 2. Mann-Whitney U test—Variables and ROIs.**

| | Null hiphotesis | Significance | Decision |
|---|---|---|---|
| 1 | The Eye Left distribution is the same across the Sex categories. | .002* | Reject the null hypothesis |
| 2 | The Eye Right distribution is the same across the Sex categories. | .030* | Reject the null hypothesis |
| 3 | The Average EYE distribution is the same across the Sex categories. | .009* | Reject the null hypothesis |
| 4 | The distribution of DF1R_0 is the same among the Sex categories. | .001* | Reject the null hypothesis |
| 5 | The distribution of DF2R_0 is the same among the Sex categories. | .001* | Reject the null hypothesis |
| 6 | The distribution of DF3R_0 is the same among the Sex categories. | .003* | Reject the null hypothesis |

*middle*, *4 ring and 5 little finger*), *R (right)*, *L(left)* and the number at the end represents time. For example, DF1R_0 refers to the dorsal part of the thumb, of the right hand at 0 minutes. According to Fig 3 and p-values associated to variables, all of them were statistically significant at 95% of confidence level. For a detailed result about this analysis and estimates, please see S3 Annexure.

**3.1.4 Friedman test between the time intervals of the experimental analysis of the ROIs.** To check if there is any difference between the recovery time intervals of the temperature ROIs for the different times considered, the Friedman test is applied. This test is performed to compare each of the ROIs of the thumb, index, middle, ring and little fingers at 0, 10, 15 and 20 minutes in both areas: dorsal and palmar area. The results obtained are shown in Table 3. The results show asymptotic significance contrasted with a significance level of 0.05. With this result, it is shown that, PF1L_0,10,15,20 (variables corresponding to the part of the palm of the left hand on the thumb, at 0, 10, 15 and 20 min.), have a difference between the recovery time intervals and this variable is highlighted in bold type. The rest of the variables do not have enough statistical evidence to prove that the medians of the tests are different.

The mean of the temperatures obtained from the dorsal and palmar part of the hand are shown in Table 4, where an increase at temperature is observed after having carried the process experiment to 10, 15, and 20 minutes of testing. The highest average temperatures were recorded in the left hand (non-dominant in most participants and in the first 3 fingers) as appears in Fig 4. The areas that were delimited in each of the fingers are the dorsal and palm areas.

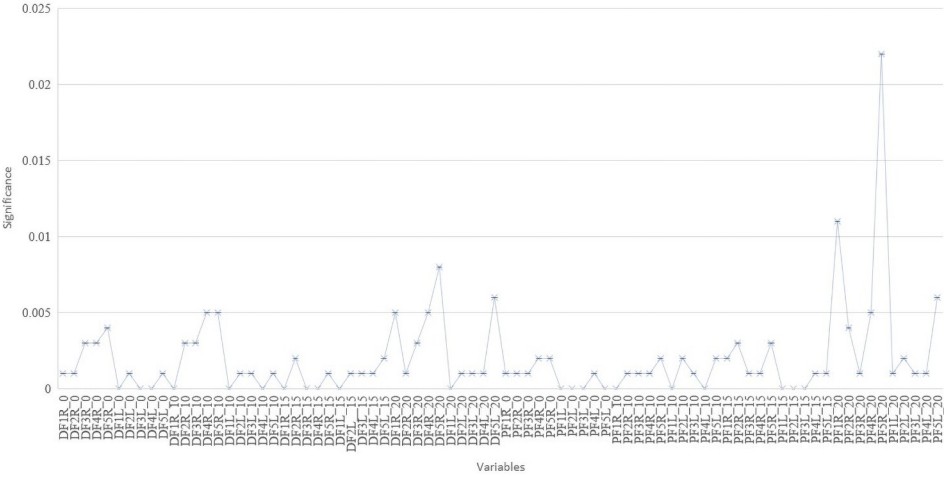

**Fig 3. Two-tailed hypothesis test at 95% confidence level for ROIs variables.**

**Table 3. Friedman test between times of ROIs.**

| Analyzed variables for each finger at 10, 15 and 20 min. | Asintotic significance |
|---|---|
| DF1R_0,10,15,20 | 0.052 |
| DF2R_0,10,15,20 | 0.064 |
| DF3R_0,10,15,20 | 0.649 |
| DF4R_0,10,15,20 | 0.878 |
| DF5R_0,10,15,20 | 0.895 |
| DF1L_0,10,15,20 | 0.041 |
| DF2L_0,10,15,20 | 0.040 |
| DF3L_0,10,15,20 | 0.103 |
| DF4L_0,10,15,20 | 0.456 |
| DF5L_0,10,15,20 | 0.690 |
| PF1R_0,10,15,20 | 0.033 |
| PF2R_0,10,15,20 | 0.190 |
| PF3R_0,10,15,20 | 0.496 |
| PF4R_0,10,15,20 | 0.731 |
| PF5R_0,10,15,20 | 0.875 |
| **PF1L**_0,10,15,20 | **0.004**[*] |
| PF2L_0,10,15,20 | 0.016 |
| PF3L_0,10,15,20 | 0.193 |
| PF4L_0,10,15,20 | 0.960 |
| PF5F_0,10,15,20 | 0.979 |

**Table 4. Average temperatures of the finger, dorsal and palmar regions.**

| | | Dorsal | | | | Palms | | | |
|---|---|---|---|---|---|---|---|---|---|
| | | Left | | Right | | Left | | Right | |
| Time (min.) | Finger | Mean | SD | Mean | SD | Mean | SD | Mean | SD |
| **Baseline** | 1 | 25.87 | 3.22 | 26.28 | 3.16 | 25.67 | 2.98 | 24.99 | 2.91 |
| | 2 | 25.05 | 3.51 | 25.04 | 3.44 | 25.09 | 3.35 | 24.86 | 3.29 |
| | 3 | 25.09 | 3.54 | 24.97 | 3.46 | 25.11 | 3.26 | 25.17 | 3.31 |
| | 4 | 24.85 | 3.46 | 24.68 | 3.39 | 24.86 | 3.21 | 24.96 | 3.32 |
| | 5 | 24.49 | 3.40 | 24.13 | 3.35 | 24.46 | 3.23 | 24.58 | 3.32 |
| **10** | 1 | 26.89 | 3.50 | 27.09 | 3.44 | 26.75 | 3.25 | 25.60 | 3.37 |
| | 2 | 26.17 | 3.70 | 25.89 | 3.80 | 26.14 | 3.55 | 25.69 | 3.57 |
| | 3 | 25.89 | 3.80 | 25.56 | 3.75 | 25.68 | 3.54 | 25.66 | 3.62 |
| | 4 | 25.45 | 3.89 | 25.11 | 3.69 | 25.19 | 3.62 | 25.46 | 3.71 |
| | 5 | 24.94 | 3.90 | 24.41 | 3.53 | 24.71 | 3.54 | 24.98 | 3.71 |
| **15** | 1 | 26.92 | 3.17 | 27.18 | 3.22 | 26.85 | 2.91 | 25.84 | 3.16 |
| | 2 | 26.10 | 3.46 | 26.02 | 3.49 | 26.06 | 3.45 | 25.79 | 3.32 |
| | 3 | 25.65 | 3.62 | 25.53 | 3.60 | 25.62 | 3.49 | 25.60 | 3.52 |
| | 4 | 25.34 | 3.73 | 25.24 | 3.59 | 25.17 | 3.55 | 25.37 | 3.62 |
| | 5 | 24.81 | 3.60 | 24.64 | 3.39 | 24.69 | 3.53 | 25.01 | 3.53 |
| **20** | 1 | 26.81 | 2.90 | 27.10 | 3.07 | 26.75 | 2.67 | 25.67 | 2.86 |
| | 2 | 26.02 | 3.26 | 25.93 | 3.25 | 25.93 | 3.26 | 25.60 | 3.13 |
| | 3 | 25.68 | 3.40 | 25.46 | 3.35 | 25.56 | 3.26 | 25.61 | 3.31 |
| | 4 | 25.38 | 3.52 | 25.14 | 3.35 | 25.19 | 3.35 | 25.48 | 3.39 |
| | 5 | 24.91 | 3.39 | 24.53 | 3.06 | 24.70 | 3.30 | 25.10 | 3.27 |

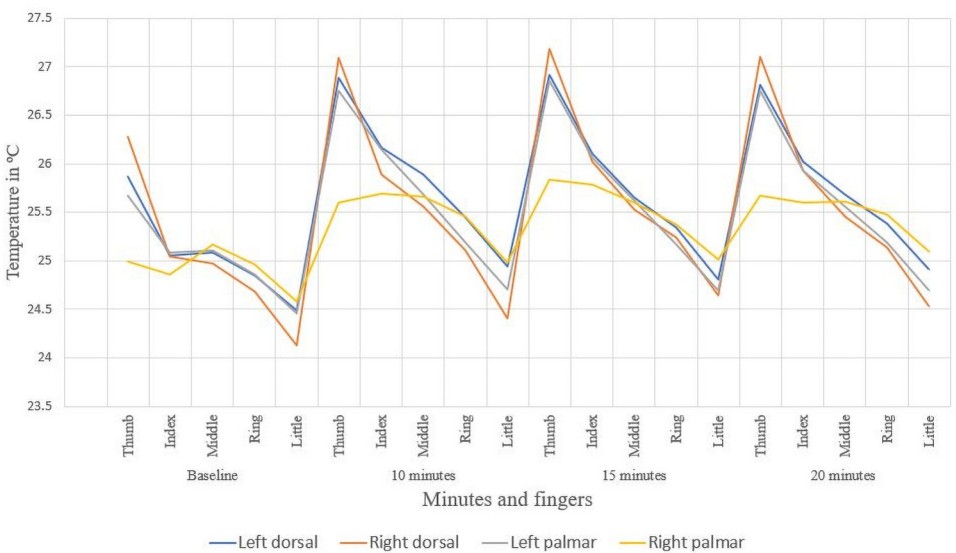

**Fig 4. Temperature of the fingers in the palmar hand.**

From the right dorsal region of the thumb, the maximum temperature increase was 0.9˚C and was recorded at 15 and 20 minutes; from the left hand, the maximum temperature increase of the thumb was 1.87˚C, at 15 minutes. The initial temperature differences of 10, 15, and 20 minutes as can be seen in Table 5. The largest temperature gradients were found in the first three fingers, thumb, index, and middle.

In Fig 5 appears the average of the right hand of each ROI in fingers, for 10, 15, and 20 minutes after the activity, while Fig 6 illustrates the case for left hand. The thumb is the one that registers the highest temperature index, unlike the rest, the behavior of the averages of both hands is very similar.

Regarding the temperature differences of the palms of the hand, the largest were those of the left palm, the first 4 fingers, and the largest was recorded from the thumb at 15 minutes with a difference of 1.17 ˚C regarding temperature initial as shown in Table 6.

Unlike the temperatures of the hand region, the palm regions differ more markedly and can be seen in Fig 7 for righ hand and Fig 8 for left hand.

Regarding the influence of anthropometric parameters on thermal results, there was statistical evidence ($p < 0.05$) on the effect of sex in most ROIs of the mean temperature at the recording times (excluding the right hand and forearm right at the baseline) but not in bilateral disputes. The practice of sports showed statistical evidence on the bilateral differences between the ROIs studied, but not in the mean temperature of the ROIs. The BMI and the history of a fracture did not present any statistical evidence of influence on the measured temperatures, this variable was divided into two groups, by sex. The correlations between the temperatures

**Table 5. Temperature differences (˚C) of the dorsal part of the hand.**

| | Fingers left hand | | | | | Fingers right hand | | | | |
|---|---|---|---|---|---|---|---|---|---|---|
| | Thumb | Index | Middle | Ring | Little | Thumb | Index | Middle | Ring | Little |
| 10 min. | 1.02 | 1.12 | 0.81 | 0.59 | 0.45 | 0.81 | 0.86 | 0.59 | 0.42 | 0.28 |
| 15 min. | 1.87 | 1.05 | 0.57 | 0.48 | 0.33 | 0.90 | 0.98 | 0.55 | 0.55 | 0.52 |
| 20 min. | 0.94 | 0.97 | 0.59 | 0.53 | 0.43 | 0.82 | 0.89 | 0.49 | 0.45 | 0.41 |

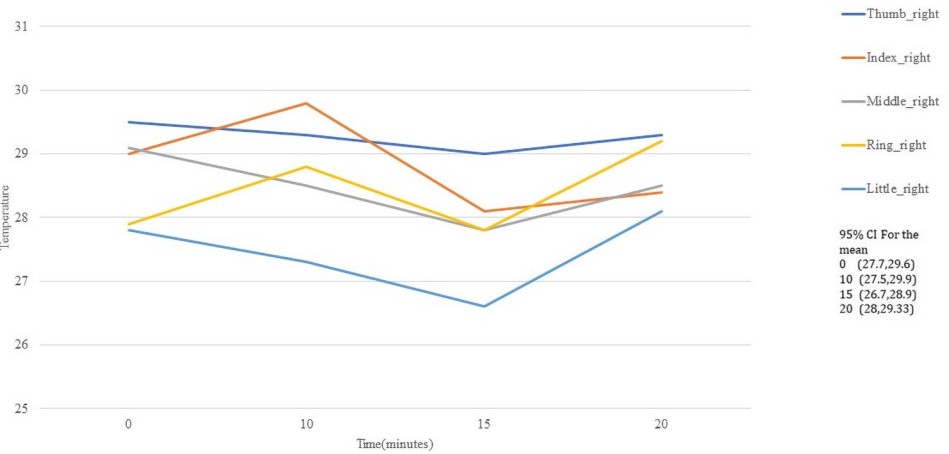

**Fig 5. Temperature behaviour of the fingers in dorsal right hand.**

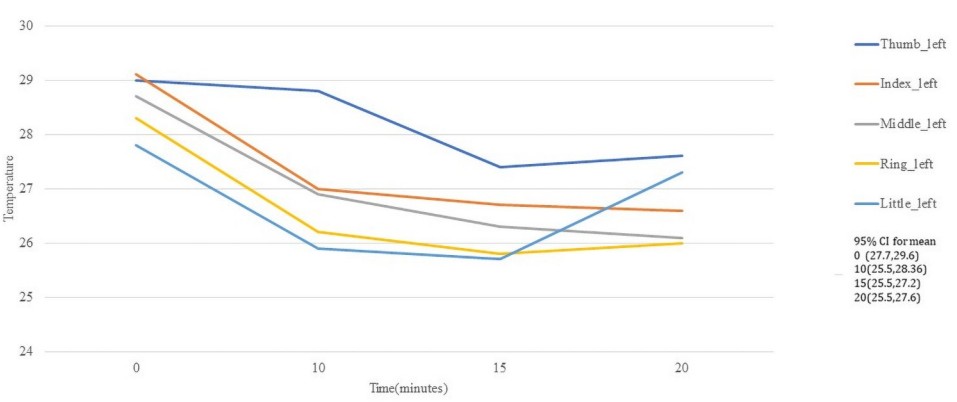

**Fig 6. Temperature behaviour of the fingers in dorsal left hand.**

and the anthropometric parameters sex, age, BMI, history of fractures, and sports habits were analyzed, those that were above 80% are the temperatures between the regions of the fingers, and none were related to the anthropometric parameters.

## 4. Discussion

This experimental study consisted of repetitive work activity with both hands, one of the main findings was that the thumb, index, and middle fingers raised the temperature more compared to the rest. A maximum temperature difference concerning the initial temperature of 1.17 °C was obtained in the palmar area for the thumb of the left hand and the dorsal area of 1.87 °C

**Table 6. Temperature differences (°C) of the palmar part of the hand.**

| | Fingers left hand(palms) | | | | | Fingers right hand (palms) | | | | |
|---|---|---|---|---|---|---|---|---|---|---|
| | Thumb | Index | Middle | Ring | Little | Thumb | Index | Middle | Ring | Little |
| **10 min.** | 1.07 | 1.05 | 0.57 | 0.33 | 0.25 | 0.61 | 0.84 | 0.49 | 0.50 | 0.40 |
| **15 min.** | 1.17 | 0.96 | 0.51 | 0.30 | 0.23 | 0.86 | 0.93 | 0.43 | 0.41 | 0.43 |
| **20 min.** | 1.08 | 0.83 | 0.45 | 0.33 | 0.25 | 0.68 | 0.74 | 0.44 | 0.52 | 0.52 |

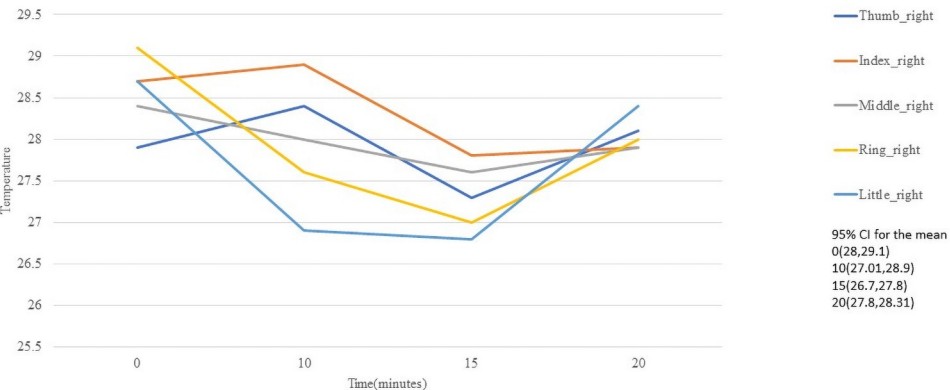

**Fig 7. Temperature behaviour of the fingers in palmar right hand.**

for the thumb; both after 15 minutes of the test. This work agrees with Ammer and Ring [42], where they take thermographic images of the dorsal hands, and conclude that to minimize the variation in ROIs, this will depend on the variation of the body position. In another study of Camargo, Ordorica [11], who in 2012 report the behavior of the temperature when performing a repetitive task for two hours, recording maximum differences in temperatures of 3.44 ˚C and 3.67 ˚C for right and left wrist respectively. In conclusion, this study shows that a longer time performing repetitive activity could increase the temperature gradient for the extremities involved.

On the other hand, the temperature gradients in this investigation were maintained above 1˚C for the first and second fingers at 10 and 15 minutes of rest after having performed the repetitive activity for 10 minutes. A difference of 1˚C from the normal temperature in 15-minute intervals may indicate a pathological problem [43] and maintaining this temperature gradient is a risk indicator.

When there is an inflammation or infection, the temperature increases, and the increase in temperature in certain areas is due to the stress induced by a repetitive task as claimed by Ramos, dos Reis [33] and Rossignoli, Benito [44]. Also Tchou, Costich [45] diagnosed CTS when there was an abnormality in which more than 25% of the measured area showed an increase in temperature of at least 1 ˚C compared to the symptom-free hand. Our study is comparable with previous authors, since they analyze the increase in temperature in certain

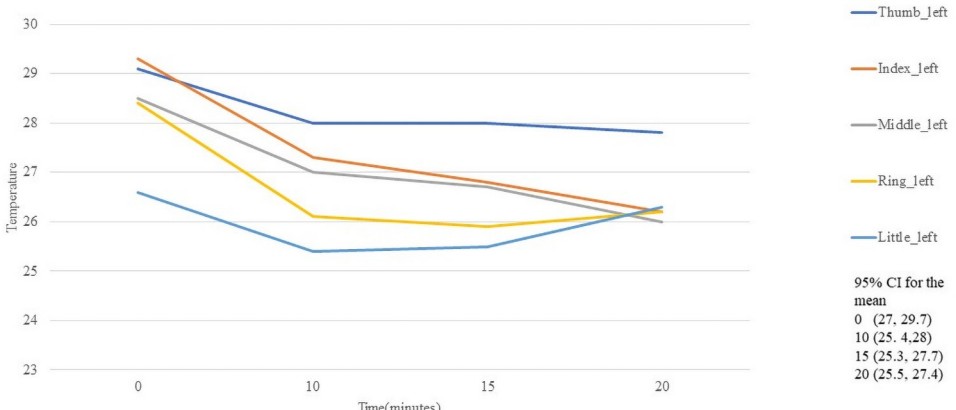

**Fig 8. Temperature behaviour of the fingers in palmar left hand.**

areas due to the stress induced by a repetitive task. Another indicator is the recovery time, for example, Horikoshi, Inokuma [46] performed a water immersion test for healthy subjects and subjects with Raynaud's Phenomenon and found that healthy subjects regained their temperature within 10 minutes and were not affected.

Repetitive tasks in the workplace result in the development of fatigue and reduced physical capacity [47]. In our findings no evidence was found to support that the increase in temperature in ROIs is correlated with fatigue, unlike the work presented by McDonald, Mulla [47] reports that after repetitive work, physical capacity is reduced.

Also, a study of the region of the palm and the back of the hand was carried out. Differences in the palm region are lower than in the dorsal part of the hand, the most significant difference is for the thumb at 15 min; maintaining a difference of 0.7 ˚C, and no significant changes were found between both areas. However, Sousa, Vardasca [48] mentioned that depending on the ROIs, a pathology of the upper extremities can be identified with greater precision.

Likewise, in our study, it was considered that some anthropometric parameters affect temperature such as age, BMI, sex, dominant hand, fractures, and whether, practice sports. However, there was statistical evidence that sex parameter influences the temperature in fingers; unlike the recovery of the dominant hand with age, because there is no statistical evidence to states a relationship with the recovery of temperature. Our study differ with Katić, Li [49], who studied regions where obese or high BMI people had a higher hand skin temperature compared to people of normal weight or BMI.

Also Horikoshi, Inokuma [46] and Fernández-Cuevas, Marins [50] had reported that MSD also influences the temperature behavior, as well as other variables such as sex, because they show that women maintain a higher temperature, among other intrinsic variables. Finally, Cazares-Manríquez, Wilson [51] mentions that in a group of people with CTS, BMI and anthropometric measurements are highly significant variables for the diagnosis of this disease.

## 5. Conclusions

Through the execution of repetitive work with all the fingers, the thermal behavior evaluated in ˚C, and the recovery of temperature in ROIs in the region of the hand, dorsal part, palmar and each one of the regions of the fingers were evaluated. It is concluded that performing repetitive work with all the fingers of the hand does not show that they all present the same temperature rise and recovery behavior, besides, not all the variables studied affect the recovery of temperature and its behavior. It was determined which fingers maintain a higher temperature when performing repetitive work, reel cables continuously these being the first three fingers: thumb, index, and middle, and the temperature recovery was achieved after 10 minutes of rest once the task was finished repetitive. Also, the sex parameter has a direct relationship to the increase in skin temperature, and that age significantly influences the recovery or stabilization of temperature.

The statistical analysis of the anthropometric variables and their correlation leads us to propose a study with a focus on different characteristics. Future research proposes to carry out a comparative analysis between people who have some MSD and healthy people, as well as to analyze the difference in behavior between genders (male—female), also, to analyze if there is a correlation between fatigue and reduction of physical capacity concerning temperature.

## Supporting information

**S1 Annexure. Nomenclature of variables.**
(DOCX)

**S2 Annexure. Results of the normality test Anderson Darling.**
(DOCX)

**S3 Annexure. Mann-Whitney U test between ROIs and the sex variable.**
(DOCX)

## Author Contributions

**Conceptualization:** Alejandra García Becerra, Claudia Camargo Wilson.

**Data curation:** Jesús Everardo Olguín-Tiznado, Jorge Luis García Alcaraz.

**Formal analysis:** Blanca Rosa García-Rivera, Ricardo Vardasca.

**Investigation:** Alejandra García Becerra, Jesús Everardo Olguín-Tiznado.

**Methodology:** Ricardo Vardasca, Juan Andres López-Barreras.

**Resources:** Alejandra García Becerra, Claudia Camargo Wilson.

**Writing – original draft:** Jesús Everardo Olguín-Tiznado, Jorge Luis García Alcaraz.

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
