## [Decision Letter · Decision Letter 0]

29 Dec 2020

PONE-D-20-36722

Infrared thermal imaging monitoring on hands when performing repetitive tasks: an experimental study

PLOS ONE

Dear Dr. Garcia Alcaraz,

Thank you for submitting your manuscript to PLOS ONE. After careful consideration, we feel that it has merit but does not fully meet PLOS ONE’s publication criteria as it currently stands. Therefore, we invite you to submit a revised version of the manuscript that addresses the points raised during the review process.

Both reviewers raised significant concerns about data analysis and task validity. If the authors wish to publish this paper on PLOS ONE, they must provide a more detailed description of the task, and find a way to address the reviewers' concerns about its standardization and external validity.

In addition to the reviewers' concerns, the authors should also address the possibility of skin tone/color affecting infrared camera accuracy.

We look forward to receiving your revised manuscript.

Kind regards,

Benjamin A. Philip

Academic Editor

PLOS ONE

Journal Requirements:

2. In the methods section please provide additional information regarding participant recruitment, in particular: a) the recruitment date range (month and year), b) a description of any inclusion/exclusion criteria that were applied to participant recruitment, c) a statement as to whether your sample can be considered representative of a larger population, d) a description of how participants were recruited, e) descriptions of where participants were recruited and where the research took place.

Furthermore, please clarify whether you have written consent for publication for the participant’s picture in figure 2. For further information please refer to our policy on informed consent for publication https://journals.plos.org/plosone/s/human-subjects-research#loc-Patient-Privacy-and-Informed-Consent-for-Publication;
https://journals.plos.org/plosone/s/file?id=8ce6/plos-consent-form-english.pdf

3.Thank you for stating the following financial disclosure:

Reviewers' comments:

Reviewer's Responses to Questions

**Comments to the Author**

1. Is the manuscript technically sound, and do the data support the conclusions?

Reviewer #1: No

Reviewer #2: Yes

2. Has the statistical analysis been performed appropriately and rigorously? 

Reviewer #1: I Don't Know

Reviewer #2: No

3. Have the authors made all data underlying the findings in their manuscript fully available?

Reviewer #1: No

Reviewer #2: Yes

4. Is the manuscript presented in an intelligible fashion and written in standard English?

Reviewer #1: Yes

Reviewer #2: Yes

5. Review Comments to the Author

Reviewer #1: The submitted manuscript provides mostly a descriptive analysis of the changes in hand skin surface radiation measured by IRT upon a 10 min. non standardized manual task.

Major aspects:

Basic aspects of data analysis are not clear.

The wire task applied is non-standardized and does not seem to involve the two small fingers of the hand.

The researchers do not report on the dominant hand, even though results seem to indicate a bit a more pronounced recovery from min. 15-20 on the right hand side.

The discussion of results and conclusions, but also large parts of the introduction refers to MSD. However, external validity of the task performed and the temperature patterns monitored for predicting succepebility for MSD.

Minor aspects:

Methodology: (Line 168 – 241)

Line 227 – 229: The measurement time points are not clearly defined. Your figures indicate other timepoints for the IRT measurement. “0” is after or before the task?

Line 232 – 234: The figure caption differs from the figure caption after the main text. Please clarify that difference.

Line 236 – 247: The statistical analysis is insufficient.

Results:

Discussion (Line 288 – 347):

Line 293 – 299: Please shorten this sentence and explain the results of the cited authors more precisely.

Line 302 – 308: Why are your results comparable to those of Nakatani et al.? Please explain more precisely. Furthermore, Nakatani was not the only author of article 38 in your references, please choose the right citation style.

Line 308 – 310: Do you think or claim here that a maintained temperature increase causes inflammation, or that inflammation leads to temperature increases in certain areas? Please verify and differentiate if you talk about acute heat increases due to acute “task induced stress” or chronic heat increases in a certain area.

Line 311: You write carpal tunnel syndrome. Before you used the abbreviation CTS. Please apply abbreviations consistently.

Line 313 – 315: Please go into detail: Which results? Cite necessary sources also in this sentence. What do you mean by slow recovery? Which time period is meant by “long time after repetitive tasks?”

Line 316 – 317: Please provide a reference.

Line 317 – 319: This sentence is hard to understand. Please rewrite it.

Line 319 – 321: Please explain the results of McDonald et al. more precisely. Furthermore, you cite “McDonald, et al” and then write: “He”… That is not feasible.

Line 322 – 324: How does this reference help to interpret your results? Please explain.

Line 325 – 328: Which reference do you relate to?

Line 328 – 331: What is more significant here?

Line 338: We think “concludes” is the wrong word here.

Line 340 – 342: Why does this citation help to interpret your data?

Line 344: You cite Fernández-Cuevas, et al. (47). They show a different outcome than you considering the correlation between sex and temperature. We recommend to mention that.

Line 345 – 347: “highly significant variables” in relation to what?

Conlusions (Line 348 – 363):

Line 349: “thermal behavior” Please name the investigated parameter.

Line 350: “temperature for different regions” Please name the ROI at least “hands”

Line 353: “risk of injury” Now you talk about risk injury, but you have not discussed about risk injury in your discussion.

Line 353 -356: Why do you write “Likewise”. Moreover, your specific conducted repetitive task leads to higher temperature in certain ROIs. Please name the specific task.

Line 357: You state sex has a direct relationship but in line 335 – 336 you described to identified no significant relationship between sex and temperature.

Line 360 – 362: How do you derive the suggestions for further research from your investigation and results?

Figures

Line 65 to 68: Your figure captions are to short and do not describe the content of the figures precisely.

Figure 2: Please provide the exact ROI that you analysed within the figure

Figures 3 &4: Please provide confidence intervals

Reviewer #2: The paper entitled “Infrared thermal imaging monitoring on hands when performing repetitive tasks: an experimental study” reports an interesting research aimed to the thermal characterization of hands during a particular case of work activities. Neither the methodology is of course a novelty nor the type of investigation. hand temperature measurements, which as reported by authors was already investigated in previous studies. Thermal images of hands were acquired pre and post-activities paying particular attention as it is requested in standard measurement protocols: thermal stabilization of the subjects, thermal condition of the environment avoiding effect of external thermal influencer.

In my personal opinion the most interesting aspect of this study is the target of the investigated subjects, workers performing repetitive tasks, that I consider of social relevance since IRT is a cheap, easy to perform and reliable technique. Considering that in these cases, more often than not, we have important consequences deriving from the work-related stress that particularly affect these type of workers.

After these general considerations I suggest to publish this research after improvement mainly related to the statistical analysis that is not specific.

In these research work, beside the interesting application, the main part to handle is the statistical analysis that in this work I consider weak as explained below.

Authors should in fact specify which test was used for each variable. For instance, Kruskal Wallis (non-parametric equivalent of the ANOVA) to evaluate if there are differences between fingers in different time intervals considered. In addition to check if there is any difference in temperature values for the different considered time, the best test for repeated measures is the ANOVA (normal distribution, parametric) or Friedman (non-parametric t-test equivalent for independent subjects). Authors considered also Mann Whitney but it is not clear what is the variable considered.

Other question: what are the normal variables?

I suggest additionally to study the effect of sex in each ROI, using an indipendent sample t-test (or non-parametric equivalent, depending if data are “normal”) and the comparison on male and female where for each ROI there is a p-value.

Summarizing is not clear what test was used for and authors should specify if it was used the Pearson’s r (parametric) or Spearman’s r (non-parametric) in the statistical analysis.

Below some specific comments:

-L92: replace gradient with distribution: we have always distribution but not gradient

-L104: ref. 18 is more related to method of temperature extraction, pleas clarify. It would be intesting to use the T-Max method in the data extraction

-L108: I suggest to add “of possible MSD” after preliminary diagnosis

-L110: replace measure instead of analyze, IRT measures, researchers do the analysis of data

-L111: skin temperature is affected not only by blood perfusion, please correct and add references

-L116, L117, L174, L250, L252, and other lines to be checked: correct temperature unit in “°C” not “° C” adding a space between data and unit (e.g. 20 °C)

-L119 what is the prolonged light? Are the findings of reference 24 related to the effect of temperature in the consider aim of the work?

-L124-135 main work deal with typing and similar activities, do the authors considered fingers friction in the analyzed repetitive work, it is very different than typing

-L244 it would be interesting to know what is ROI pixel area to check if it can be statistically meaningful for the used data

-L252 replace gradient with differences

-Discussion section: in this section, authors report the possible relation of temperature increase/decrease of the considered body area with the risk of injuries. These speculations where authors report previous studies can or cannot be related to this specific research finding and I suggest to move this part in the Introduction. I think that discussions need to be related to the direct finding of this research, possibly with the statistical results correlated with sex, age, BMI, and other external variable that could have influenced the hands temperature values.

6. PLOS authors have the option to publish the peer review history of their article (what does this mean?). If published, this will include your full peer review and any attached files.

Reviewer #1: No

Reviewer #2: No

---

## [Author Response · Author response to Decision Letter 0]

20 Feb 2021

RESPONSE TO EDITOR

Dear Editor, the authors appreciate your comments and suggestions for improve the paper. Next paragraphs illustrate in bold letters your comments and our response appear in italic.

Editor. Thank you for submitting your manuscript to PLOS ONE. After careful consideration, we feel that it has merit but does not fully meet PLOS ONE’s publication criteria as it currently stands. Therefore, we invite you to submit a revised version of the manuscript that addresses the points raised during the review process.

Our response: The authors are grateful for your comments and would like to inform you that the article has been restructured and modified with the suggested adjustments.

Editor. Both reviewers raised significant concerns about data analysis and task validity. If the authors wish to publish this paper on PLOS ONE, they must provide a more detailed description of the task, and find a way to address the reviewers' concerns about its standardization and external validity.

Our response. A more detailed form and explanation of the task is restated:

Lines 231-233: 

Emulated tests of a work task were carried out in a cable assembly area in a harness assembly company for the automotive industry, the activity was emulated in a laboratory under conditions in which it would work in real life.

Lines 234-236:

The process consisted of rolling a pair of wires by hand for 10 minutes. This movement involves the 10 fingers of the hands and twists of the wrists, not all fingers work in the same way, however, they do have movement when performing the task. 

Lines 241-244

Minute 0 is considered before the start of the repetitive movement and after 10 minutes of activity with repetitive movement, 3 pictures of thermographic images were taken at intervals of 10, 15, and 20 minutes. The hands are placed on a template, maintaining the same posture in each shot.

Lines 264-271, its normalization and external.

Editor. In addition to the reviewers' concerns, the authors should also address the possibility of skin tone/color affecting infrared camera accuracy.

Our response. Lines 111-114 a paragraph was added to prove that skin tone and accuracy are not significantly affected.

Human skin has an emissivity of 0.98 and Charlton, Stanley (22) tested 65 people to see if skin color affected thermography precision results, and found that this was not significant, in other words, skin pigmentation does not affect the thermal emissivity measurement.

R= Item meets PLOS ONE shape and style template requirement

2. In the methods section please provide additional information regarding participant recruitment, in particular: a) the recruitment date range (month and year),

R= This is described on line 197-198. "They were recruited in the month of October and November 2017 in a laboratory of the Autonomous University of Baja California (AUBC)"

b) a description of any inclusion/exclusion criteria that were applied to participant recruitment,

R= This is described in lines 207-210:

The exclusion criteria for participants were that they should not have consumed caffeine, drank alcohol, or exercised prior to the test. Before the experimental tests, participants were asked not to drink alcohol, tea, coffee, or smoke or use any ointment on their skin

And lines 216-220:

The recruitment of the participants took place at the AUBC with undergraduate and graduate students and some university workers. The objective of the experiment was explained to them, and they were summoned to carry out the tests later. The sample is considered representative of a larger population since it includes both genders between an age range of 19 to 38 years, including a pregnant woman. 

c) a statement as to whether your sample can be considered representative of a larger population,

R= Lines 218-220: 

The sample is considered representative of a larger population since it includes both genders between an age range of 19 to 38 years, including a pregnant woman.

d) a description of how participants were recruited

R= Lines 216-218:

The recruitment of the participants took place at the AUBC with undergraduate and graduate students and some university workers. The objective of the experiment was explained to them, and they were summoned to carry out the tests later. 

e) descriptions of where participants were recruited and where the research took place.

R= This is described on lines 216-218

The recruitment of the participants took place at the AUBC with undergraduate and graduate students and some university workers. The objective of the experiment was explained to them, and they were summoned to carry out the tests later. 

Furthermore, please clarify whether you have written consent for publication for the participant’s picture in figure 2.

R= You have the consent of this photo in writing and even so it is clarified within the document

For further information please refer to our policy on informed consent for publication https://journals.plos.org/plosone/s/human-subjects-research#loc-Patient-Privacy-and-Informed-Consent-for-Publication;
https://journals.plos.org/plosone/s/file?id=8ce6/plos-consent-form-english.pdf

3.Thank you for stating the following financial disclosure:

R= The authors did not receive any specific funding for this work.

a. Please clarify the sources of funding (financial or material support) for your study. List the grants or organizations that supported your study, including funding received from your institution.

R= The authors did not receive any specific funding for this work.

R= Funders had no role in study design, data collection and analysis, decision to publish, or manuscript preparation.

If any authors received a salary from any of your funders, please state which authors and which funders.

R= No author received a salary.

If you did not receive any funding for this study, please state: “The authors received no specific funding for this work.”

R=” The authors did not receive any specific funding for this work. "

4. We note that you have indicated that data from this study are available upon request. PLOS only allows data to be available upon request if there are legal or ethical restrictions on sharing data publicly. For information on unacceptable data access restrictions, please see http://journals.plos.org/plosone/s/data-availability#loc-unacceptable-data-access-restrictions

R= There are no ethical or legal restrictions on sharing a de-identified data set.

R= There are no ethical or legal restrictions on sharing a de-identified data set.

If there are no restrictions, please upload the minimal anonymized data set necessary to replicate your study findings as either Supporting Information files or to a stable, public repository and provide us with the relevant URLs, DOIs, or accession numbers. Please see http://www.bmj.com/content/340/bmj.c181.long for guidelines on how to de-identify and prepare clinical data for publication. For a list of acceptable repositories, please see

http://journals.plos.org/plosone/s/data-availability#loc-recommended-repositories.

RESPONSES TO REVIEWER 1

The submitted manuscript provides mostly a descriptive analysis of the changes in hand skin surface radiation measured by IRT upon a 10 min. non standardized manual task.

Major aspects:

Dear reviewer, we appreciate your valuable time, comments and feedback to improve the quality in our manuscript. Your comment appears in black words, while our responses are in italic for a better understanding. 

Basic aspects of data analysis are not clear.

Response: We appreciate your suggestions. Page 12 to 19, Lines 253 to 305: shows the statistical analysis in section 2.4, with the results of normality tests and Friedman's test for mean temperatures, and the Mann-Whintney U-test between ROIs for each variable including temperature.

The wire task applied is non-standardized and does not seem to involve the two small fingers of the hand.

Response: We appreciate your suggestions. Page 12, line 230 to 235: “Emulated tests of a work task were carried out in a cable assembly area in a harness assembly company for the automotive industry, the activity was emulated in a laboratory under conditions in which it would work in real life.

This movement involves the movement of the 10 fingers of the hand, not all of them work in the same way, however, they all have movement when performing the task.

The researchers do not report on the dominant hand, even though results seem to indicate a bit a more pronounced recovery from min. 15-20 on the right-hand side.

Response: We appreciate your suggestions. Page 11, line 227 to 228: the dominant hand is reported, in this case all participants were right-handed.

The discussion of results and conclusions, but also large parts of the introduction refers to MSD. However, external validity of the task performed and the temperature patterns monitored for predicting succepebility for MSD.

Response: A description was added showing the difference of both terms on line 95-102 both terms are explained and the importance of including the MSD.

MSD is a term that groups together diseases that affect different parts of the locomotor system caused by its excessive use (17), caused by prolonged and tiring work and postures, improper unbalanced angles, load handling and repetitive movements (18). DUE are accumulated lesions that frequently occur in connective soft tissues, particularly to tendons and their sheaths. An MSD can irritate or damage nerves and impede blood flow through arteries and veins. They are recurrent in the wrist area, shoulder, and neck. For example, Carpal Tunnel Syndrome (CTS), tendinitis, tendosynovitis, Guyon's tunnel syndrome, are diseases resulting from repetitive efforts within the work occupation (19). And the term DUE was modified by MSD in the lines 152,163,164,414.

Minor aspects:

Methodology: (Line 168 – 241)

Line 227 – 229: The measurement time points are not clearly defined. Your figures indicate other timepoints for the IRT measurement. “0” is after or before the task?

Response: The initial photos were called minute 0 before starting the repetitive movement is analyzed in line 240

Line 232 – 234: The figure caption differs from the figure caption after the main text. Please clarify that difference.

Response: The description of figure 2 is included in line 250-252. The legend is clarified in the main part of the text, and a correction is made in the description of the image.

Line 236 – 247: The statistical analysis is insufficient.

Response: In this part, normality tests were included, statistical analysis were included in the statistical analysis part, including normality tests and the Friedman test for mean temperatures, and a Mann whintney u test between ROIs, for each variable that included temperature 253-305. 

Results:

Discussion (Line 288 – 347):

Line 293 – 299: Please shorten this sentence and explain the results of the cited authors more precisely.

Response: In this paragraph, the adjustments were made specifying the results of the two cited authors. The text was as follows:

This experimental study consisted of repetitive work activity with both hands, one of the main findings was that the thumb, index, and middle fingers raised the temperature more compared to the rest. A maximum temperature difference concerning the initial temperature of 1.17 °C was obtained in the palmar area for the thumb of the left hand and the dorsal area of 1.87 °C for the thumb; both after 15 minutes of the test. This work agrees with Ammer and Ring (41),where he takes thermographic images of the dorsal hands, and concludes to minimize the variation in ROIs, this will depend on the variation of the body position. In another study of Camargo, Ordorica (11), who in 2012 the behavior of the temperature is analyzed when performing a repetitive task for two hours, recording maximum temperatures of 3.44 ºC and 3.67 ºC for right and left wrist respectively. In conclusion, this study shows that a longer time performing repetitive activity could increase the temperature gradient for the extremities involved. This is found on lines 348-359

Line 302 – 308: Why are your results comparable to those of Nakatani et al.? Please explain more precisely. Furthermore, Nakatani was not the only author of article 38 in your references, please choose the right citation style.

Response: The Nakatani reference was deleted because it was considered that it was not focused on the results obtained in our research. A part of the paragraph was also restructured.

On the other hand, the temperature gradients in this investigation were maintained above 1°C for the first and second fingers at 10 and 15 minutes of rest after having performed the repetitive activity for 10 minutes. In order to maintain a temperature increase by 1ºC difference from the normal temperature in 15-minute intervals may indicate a pathological problem (42). In both cases, maintaining this increase in the temperature gradient is a risk indicator. When there is an inflammation or infection, the temperature increases, and the increase in temperature in certain areas is due to the stress induced by a repetitive task as claimed Ramos, dos Reis (33) and Rossignoli, Benito (43). Also Tchou, Costich (44) diagnosed CTS when there was an abnormality in which more than 25% of the measured area showed an increase in temperature of at least 1 °C compared to the symptom-free hand.This study is comparable with previous authors, since they analyze the increase in temperature in certain areas due to the stress induced by a repetitive task. Another indicator is the recovery time, Horikoshi, Inokuma (45) performed a water immersion test for healthy subjects and subjects with Raynaud's Phenomenon and found that healthy subjects regained their temperature within 10 minutes and were not affected. Lines 360-374

Line 308 – 310: Do you think or claim here that a maintained temperature increase causes inflammation, or that inflammation leads to temperature increases in certain areas? Please verify and differentiate if you talk about acute heat increases due to acute “task induced stress” or chronic heat increases in a certain area.

Response: In lines 370-372 this observation is explained

"When there is inflammation or infection, the temperature increases, and the increase in temperature in certain areas is due to the stress induced by a repetitive task"

Line 311: You write carpal tunnel syndrome. Before you used the abbreviation CTS. Please apply abbreviations consistently.

Response: This detail was corrected in each of the parts mentioned in the CTS, line 368.

Line 313 – 315: Please go into detail: Which results? Cite necessary sources also in this sentence. What do you mean by slow recovery? Which time period is meant by “long time after repetitive tasks?”

Response: Text lines 370-374 have been adjusted. And it was as follows. 

This study is comparable with previous authors, since they analyze the increase in temperature in certain areas due to the stress induced by a repetitive task. Another indicator is the recovery time, Horikoshi, Inokuma (45) performed a water immersion test for healthy subjects and subjects with Raynaud's Phenomenon and found that healthy subjects regained their temperature within 10 minutes and were not affected.

Line 316 – 317: Please provide a reference.

Response: This reference is number 46 and was added to the text.

Line 317 – 319: This sentence is hard to understand. Please rewrite it.

Response: We add in line 375-378 “Repetitive tasks in the workplace result in the development of fatigue and reduced physical capacity (46). In the findings of this study, no evidence was found that the increase in temperature in ROIs is correlated with fatigue.Unlike the work presented by McDonald, Mulla (46) where they found that after repetitive work physical capacity is reduced”

Line 319 – 321: Please explain the results of McDonald et al. more precisely. Furthermore, you cite “McDonald, et al” and then write: “He”… That is not feasible.

Response: Corrected the way of referencing and adjusted the text. Line 375-378 “Repetitive tasks in the workplace result in the development of fatigue and reduced physical capacity (46). In the findings of this study, no evidence was found that the increase in temperature in ROIs is correlated with fatigue.Unlike the work presented by McDonald, Mulla (46) where they found that after repetitive work physical capacity is reduced”

Line 322 – 324: How does this reference help to interpret your results? Please explain.

Response: It was considered moving this reference to the introduction as it was not focused on the discussion of results. This reference was moved to the introduction. 143-145

Line 325 – 328: Which reference do you relate to?

Response: With the reference of Sousa ref. 47 and the regions that the study.

Line 328 – 331: What is more significant here?

Response: Line 382-383, mentioned depending on the ROIs, a pathology of the upper extremities can be identified with greater precision

Line 338: We think “concludes” is the wrong word here.

Response: The word was changed to " differ line 390”

Line 340 – 342: Why does this citation help to interpret your data?

Response: This quote was omitted. It is considered that it does not have a concrete approach with the findings of our study.

Line 344: You cite Fernández-Cuevas, et al. (47). They show a different outcome than you considering the correlation between sex and temperature. We recommend to mention that.

Response: Sex and temperature are variables that have a correlation in our study. This part was modified. Line 395

Line 345 – 347: “highly significant variables” in relation to what?

Response: Highly significant with the diagnosis of CTS. Line 399.

Conlusions (Line 348 – 363):

Line 349: “thermal behavior” Please name the investigated parameter.

Response: In Degrees Centigrade. Line 402

Line 350: “temperature for different regions” Please name the ROI at least “hands”

Response: Through the execution of repetitive work with all the fingers, the thermal behavior evaluated in °C, and the recovery of temperature in ROIs in the region of the hand, dorsal part, palmar and each one of the regions of the fingers were evaluated.401-403

Line 353: “risk of injury” Now you talk about risk injury, but you have not discussed about risk injury in your discussion.

Response: This part was adapted on lines 403-406

It is concluded that performing repetitive work with all the fingers of the hand does not show that they all present the same temperature rise and recovery behavior, besides, not all the variables studied affect the recovery of temperature and its behavior

Line 353 -356: Why do you write “Likewise”. Moreover, your specific conducted repetitive task leads to higher temperature in certain ROIs. Please name the specific task.

Response: This part was adapted on lines 406-410

It was determined which fingers maintain a higher temperature when performing repetitive work, (reeling cables continuously). The first three fingers maintained the highest temperature: thumb, index, and middle, and the temperature recovery was achieved after 10 minutes of rest once the task was finished repetitive.

Line 357: You state sex has a direct relationship but in line 335 – 336 you described to identified no significant relationship between sex and temperature.

Response: This part was corrected since in our results sex does directly influence the temperature variables.

Line 360 – 362: How do you derive the suggestions for further research from your investigation and results?

Response: From the statistical analysis of the anthropometric variables and their correlation leads us to propose a greater analysis with different characteristics, this is found in lines 413-414

Figures

Line 65 to 68: Your figure captions are to short and do not describe the content of the figures precisely.

Response: The captions for your figures are too short and their content is not accurately described.

The text was changed and it was like this: Figure 1 shows in steps the activity that the participants carried out, the process consisted of winding cables without stopping for 10 minutes, L245-246.

Figure 2: Please provide the exact ROI that you analysed within the figure

Response: It was corrected, and it looked like this: The ROIs that were considered in the face were the carotids, which are found in the tear duct of the eye, they were delimited and in the hands each of the regions of the fingers were delimited, Lines 250-252.

Figures 3 &4: Please provide confidence intervals

Response: The ranges were provided within the figure.

RESPONSES TO REVIEWER 2

Response: Dear reviewer, we appreciate your valuable time, comments and feedback to improve the quality in our manuscript. Your comment appears in black words, while our responses are in italic for a better understanding. 

The paper entitled “Infrared thermal imaging monitoring on hands when performing repetitive tasks: an experimental study” reports an interesting research aimed to the thermal characterization of hands during a particular case of work activities. Neither the methodology is of course a novelty nor the type of investigation. hand temperature measurements, which as reported by authors was already investigated in previous studies. Thermal images of hands were acquired pre and post-activities paying particular attention as it is requested in standard measurement protocols: thermal stabilization of the subjects, thermal condition of the environment avoiding effect of external thermal influencer.

In my personal opinion the most interesting aspect of this study is the target of the investigated subjects, workers performing repetitive tasks, that I consider of social relevance since IRT is a cheap, easy to perform and reliable technique. Considering that in these cases, more often than not, we have important consequences deriving from the work-related stress that particularly affect these type of workers.

After these general considerations I suggest to publish this research after improvement mainly related to the statistical analysis that is not specific.

This part was corrected and the tests of normality, Mann Whitney and Friedman were added to analyze the Sex variable with the ROIs.

In these research work, beside the interesting application, the main part to handle is the statistical analysis that in this work I consider weak as explained below.

Authors should in fact specify which test was used for each variable. For instance, Kruskal Wallis (non-parametric equivalent of the ANOVA) to evaluate if there are differences between fingers in different time intervals considered. In addition to check if there is any difference in temperature values for the different considered time, the best test for repeated measures is the ANOVA (normal distribution, parametric) or Friedman (non-parametric t-test equivalent for independent subjects).

Response: This was done and described in the line 253-305. 

In addition to checking if there is any difference in the temperature values for the different times considered, the best test for repeated measures is the ANOVA (normal distribution, parametric) or Friedman (non-parametric equivalent of the t test for independent subjects).

Response: This test was performed and the results are in the lines of 292-305

Authors considered also Mann Whitney but it is not clear what is the variable considered. Other question: what are the normal variables?

Response: This analysis was performed and is found on lines 256-271 described in Table 3.

I suggest additionally to study the effect of sex in each ROI, using an indipendent sample t-test (or non-parametric equivalent, depending if data are “normal”) and the comparison on male and female where for each ROI there is a p-value.

Response: This analysis is found in table 5 of line 285-291

Summarizing is not clear what test was used for and authors should specify if it was used the Pearson’s r (parametric) or Spearman’s r (non-parametric) in the statistical analysis.

Response: The analysis was Spearman (nonparametric) line 280-281

Below some specific comments:

-L92: replace gradient with distribution: we have always distribution but not gradient

Response: This word was replaced L88

-L104: ref. 18 is more related to method of temperature extraction, pleas clarify. It would be intesting to use the T-Max method in the data extraction

Response: It was decided to remove this reference

-L108: I suggest to add “of possible MSD” after preliminary diagnosis

Response: Suggested change made L114

-L110: replace measure instead of analyze, IRT measures, researchers do the analysis of data

Response: Suggested change made L116

-L111: skin temperature is affected not only by blood perfusion, please correct and add references

Response: In this section this part was added:

Another advantage is that the IT can measures the temperature distribution through an image and monitor the surface of interest (24). Gold et al.(25) showed that there is a moderate correlation between blood flow volume and mean skin temperature during a typing task in which the speed of activity affects both factors. Besides, skin temperature is affected by blood perfusion; therefore, diseases that affect blood circulation, especially in the extremities, can be studied with this technique (26). The hypothalamus controls body temperature and determines the value to which it will regulate (27), it balances heat generation with heat loss, is connected to the pituitary gland at the base of the brain near the termination of the brainstem (14). L115-124

-L116, L117, L174, L250, L252, and other lines to be checked: correct temperature unit in “°C” not “° C” adding a space between data and unit (e.g. 20 °C)

Response: These suggestions were corrected. L127, L128, L140, L142, L143, L223, L252, L315, L315, L331, L353, L358, L362, L370, L382

-L119 what is the prolonged light? Are the findings of reference 24 related to the effect of temperature in the consider aim of the work?

Response: Light is not related to temperature, it was referring to light prolonged work, L130-131.

-L124-135 main work deal with typing and similar activities, do the authors considered fingers friction in the analyzed repetitive work, it is very different than typing

Response: These references were removed. 

-L244 it would be interesting to know what is ROI pixel area to check if it can be statistically meaningful for the used data

Response: The areas that were delimited in each of the fingers are the dorsal area and the palm, L312.

-L252 replace gradient with differences

Response: This change was made, L312.

-Discussion section: in this section, authors report the possible relation of temperature increase/decrease of the considered body area with the risk of injuries. These speculations where authors report previous studies can or cannot be related to this specific research finding and I suggest moving this part in the Introduction

Response: Information that is considered to be part of the introduction was transferred to line 134-136. In addition, the text was adapted to contrast the results of other authors who have carried out similar studies. Studies that were carried out by means of thermography in the area of the hands.

- I think that discussions need to be related to the direct finding of this research, possibly with the statistical results correlated with sex, age, BMI, and other external variable that could have influenced the hands temperature values.

Response: Age, BMI and other external variables that could have influenced the values of the temperature of the hands. Those changes were addressed and placed on lines 388-390.

---

## [Decision Letter · Decision Letter 1]

22 Mar 2021

PONE-D-20-36722R1

Infrared thermal imaging monitoring on hands when performing repetitive tasks: an experimental study

PLOS ONE

Dear Dr. Garcia Alcaraz,

Thank you for submitting your manuscript to PLOS ONE. After careful consideration, we feel that it has merit but does not fully meet PLOS ONE’s publication criteria as it currently stands. Therefore, we invite you to submit a revised version of the manuscript that addresses the points raised during the review process.

We look forward to receiving your revised manuscript.

Kind regards,

Benjamin A. Philip

Academic Editor

PLOS ONE

Journal Requirements:

Reviewers' comments:

Reviewer's Responses to Questions

**Comments to the Author**

1. If the authors have adequately addressed your comments raised in a previous round of review and you feel that this manuscript is now acceptable for publication, you may indicate that here to bypass the “Comments to the Author” section, enter your conflict of interest statement in the “Confidential to Editor” section, and submit your "Accept" recommendation.

Reviewer #1: All comments have been addressed

Reviewer #2: (No Response)

2. Is the manuscript technically sound, and do the data support the conclusions?

Reviewer #1: Yes

Reviewer #2: Yes

3. Has the statistical analysis been performed appropriately and rigorously? 

Reviewer #1: Yes

Reviewer #2: Yes

4. Have the authors made all data underlying the findings in their manuscript fully available?

Reviewer #1: Yes

Reviewer #2: Yes

5. Is the manuscript presented in an intelligible fashion and written in standard English?

Reviewer #1: Yes

Reviewer #2: Yes

6. Review Comments to the Author

Reviewer #1: The authors have addressed all issues raised, appropriately. The manuscript has improved in clarity and a detailed dataset is available for download.

Reviewer #2: The paper was improved and authors replied to mosto of the issues raised by the reviewers.

Beside this consideration I suggest additional changes before the publication of the paper.

Authors added several huge tables without a specific reason, please remove them or move to appendix or additional materials. Two-page table is almost illegible.

-For instance, I suggest to remove table 3, or move to the additional materials, since for the comprehension of the results it is enough to explain for which variables normality was tested and then report that for normal variables it was used a certain parametric test, for those not normal it was used a non-parametric test adding which one and why.

-What happened to normal variables? I couldn’t find any consideration about them.

-I would move also tab 2 in appendix or additional materials.

-I found that authors mixed methods and results, for instance I suggest to move tab 5 with Mann Whitney in results section and I suggest also to replace the content of the table with a self-explanatory graph. But most of the data put in section 2 should be moved to section 3.

-In table 7 I also suggest to report the content of the table with a graph, for instance using x for time and y for ROIs T, one for the dorsal and the other for the palm.

-It would be useful to see an image with the ROIs positioned on the hands to see where the temperature extraction was made, adding also the average number of pixel considered in the ROI

-Table 8: replace temperature gradient with temperature differences and add if the data are statistically significant adding a “*” for those with p<0.05 or a column with p-values.

-Line 271: replace text with test

-Line 272 replace “To test the relationship between all variables” with “To test the effect of…”

-Line 273: what the authors mean with “sport” it is a very generic term.

-Line 279 add to ROIs “temperature”

-How BMI was used to divide the two groups?

7. PLOS authors have the option to publish the peer review history of their article (what does this mean?). If published, this will include your full peer review and any attached files.

Reviewer #1: No

Reviewer #2: No

---

## [Author Response · Author response to Decision Letter 1]

30 Mar 2021

Reply to the reviewer

Dear reviewer, we appreciate your comments for improve this paper. In following paragraphs, your comments are in bold and our response in italic.

The paper was improved, and authors replied to most o of the issues raised by the reviewers. Beside this consideration I suggest additional changes before the publication of the paper.

Thanks a lot for that comment. We have attended yours comments and hope to fulfill your expectations. 

Authors added several huge tables without a specific reason, please remove them or move to appendix or additional materials. Two-page table is almost illegible.

Considering your suggestion, tables were eliminated and in current version of the paper, three tables are reported as annex or supporting material. 

-For instance, I suggest removing table 3, or move to the additional materials, since for the comprehension of the results it is enough to explain for which variables normality was tested and then report that for normal variables it was used a certain parametric test, for those not normal it was used a non-parametric test adding which one and why.

R= Your suggestion was considered and table 3 was removed and appears as supporting material. 

What happened to normal variables? I couldn’t find any consideration about them.

R= They were analyzed through non-parametric tests, no special treatment was given to these variables since all the analysis was carried out together with ROIs and variables. 

-I would move also tab 2 in appendix or additional materials.

R= Table 2 was moved to supporting material 

-I found that authors mixed methods and results, for instance I suggest to move tab 5 with Mann Whitney in results section and I suggest also to replace the content of the table with a self-explanatory graph. But most of the data put in section 2 should be moved to section 3.

R=Statistical analyzes were changed to section 3 of results and referred to Annex 3 because is too long and difficult to fit in only one page.

Also in this current version, Figure 4 was added because in more self-explanatory. Figure 3 appears as follow.

-In table 7 I also suggest to report the content of the table with a graph, for instance using x for time and y for ROIs T, one for the dorsal and the other for the palm.

R=This observation was addressed and Figure 4 was added as follow. 

-It would be useful to see an image with the ROIs positioned on the hands to see where the temperature extraction was made, adding also the average number of pixel considered in the ROIs

R= In figure two, each of the ROIs can be seen delimited. The pixel of the camera are in L174, the average number pixel in ROIs is 2x3

-Table 8: replace temperature gradient with and add if the data are statistically significant adding a “*” for those or a column with p-values.

R=In table 8 temperature gradient was replaced with temperature differences and table 9, and adding * for values whit with p<0.05

-Line 271: replace text with test

R= The word text was replaced for test

-Line 272 replace “To test the relationship between all variables” with “To test the effect of…”

R=The text was replaced whit “To test the effect of…”

-Line 273: what the authors mean with “sport” it is a very generic term.

R= Thanks for that observation. The text sport was replaced whit participants that practice some sports

-Line 279 add to ROIs “temperature”

R= Thanks for that comment. That word was add 

-How BMI was used to divide the two groups?

R= The database used for the statistical analysis divided by the sex of the participants

---

## [Editor Report · Decision Letter 2]

6 Apr 2021

PONE-D-20-36722R2

Infrared thermal imaging monitoring on hands when performing repetitive tasks: an experimental study

PLOS ONE

Dear Dr. Garcia Alcaraz,

Thank you for submitting your manuscript to PLOS ONE. After careful consideration, we feel that it has merit but does not fully meet PLOS ONE’s publication criteria as it currently stands. Therefore, we invite you to submit a revised version of the manuscript that addresses the points raised during the review process.

This manuscript is nearly ready for publication, and was not sent out to reviewers. In a journal with a post-acceptance proofing process, this might be an acceptance. However, there are some issues that should be addressed before this manuscript is publication-ready.

1) Please provide (in the text) a full clarification and explanation of the reviewer's last comment: how BMI was used.

2) Please provide more detailed figure legends that allow the reader to understand the figures without needing to dig into the text. (I phrase this in terms of "figure legends" but may require alterations to the figures as well.) In general, figure legends should describe the figure's key finding/conclusion (ideally as the opening sentence of the legend). In addition, specifically:

Figure 1: The legend should contain the text in lines 246-247.

Figure 2: Legend should contain the text in lines 249-251. Should be written to clarify that the "2x3" number applies to the tear ducts, not the fingers.

Figure 3: The text on lines 281-285 does not seem to match the figure or figure legend. For this figure, it ought to indicate which points meet the significance threshold, and its legend ought to explain key abbreviations. However, my suspicion is that this figure is one that has been pushed to Supplementary Material, and that you should remove references "Figure 3" in 281-285. The real "Temperature of the fingers of the dorsal hand" figure is missing.

Figure 4: The horizontal axis should be clarified - it is unclear to me how those numbers work.

Figure 5: 5a/5b either need to be combined into a single image containing both charts, or be separated out fully into two figures (with separate numbers and legends). 

Figure 6: 6a/6b as above.

We look forward to receiving your revised manuscript.

Kind regards,

Benjamin A. Philip

Academic Editor

PLOS ONE
---

## [Author Response · Author response to Decision Letter 2]

8 Apr 2021

Response to Editor

Dear editor, we appreciate your comments and recommendations for improve our paper. In current version, the track change in word has been activated and you can review easier. 

The following paragraphs contain your comments in bold type, while our responses are in italics. Here is important to mention that all Figures has been tested and passed the Preflight Analysis and Conversion Engine (PACE) digital diagnostic. 

This manuscript is nearly ready for publication and was not sent out to reviewers. In a journal with a post-acceptance proofing process, this might be an acceptance. However, there are some issues that should be addressed before this manuscript is publication ready.

1. Please provide (in the text) a full clarification and explanation of the reviewer's last comment: how BMI was used.

Our response. Thanks a lot for that comment. After a review, we see that translator keep the acronym IMC in Spanish and in current version of the paper we change everything to BMI (Body Mass Index). Lines 276 – 277 indicates that variables as age, BMI, fractures, and sports do not have an influence on ROI temperature and that is why we do not report tables referring BMI. 

2) Please provide more detailed figure legends that allow the reader to understand the figures without needing to dig into the text. (I phrase this in terms of "figure legends" but may require alterations to the figures as well.) In general, figure legends should describe the figure's key finding/conclusion (ideally as the opening sentence of the legend). In addition, specifically:

Figure 1: The legend should contain the text in lines 246-247.

Our response. The legend for Figure 1 was replaced by: Steps of activity that participants have carried out. The process consisted of winding cables without stopping for 10 minutes.

Figure 2: Legend should contain the text in lines 249-251. Should be written to clarify that the "2x3" number applies to the tear ducts, not the fingers.

Our response. The legend for Figure 2 was replaced by: Thermogram of face and palm of the hand with their respective region of interest. The ROIs for face were the carotids, which are found in the tear duct of the eye and ROIs for hands were each region delimited by fingers. Resolution average was 160 x 120 pixels. 

Figure 3: The text on lines 281-285 does not seem to match the figure or figure legend. For this figure, it ought to indicate which points meet the significance threshold, and its legend ought to explain key abbreviations. However, my suspicion is that this figure is one that has been pushed to Supplementary Material, and that you should remove references "Figure 3" in 281-285. The real "Temperature of the fingers of the dorsal hand" figure is missing.

Our response: Thanks for that comment. In current version of the paper, it is indicated that all variables were statistically significant in line 288. The abbreviations were explained in lines 284-286.The name for Figure 3 was changed to: Two-tailed hypothesis test at 95% confidence level for ROIs variables, and now is according to the text in paragraph. 

Figure 4: The horizontal axis should be clarified - it is unclear to me how those numbers work.

Our response: Thanks a lot for that comment. Current version of Figure 4 has been adapted and x-axis was modified. 

Figure 5: 5a/5b either need to be combined into a single image containing both charts or be separated out fully into two figures (with separate numbers and legends). 

Our response: Thanks a lot for that comment. Figure 5a now is Figure 5 and Figure 5b now is Figure 6, and both have different legend. The call-in text also was modified.

Figure 6: 6a/6b as above.

Our response: The graphs were separated. Figure 6a in current version is Figure 7 and Figure 6b is now Figure 8. The call-in text also was modified.

---

## [Editor Report · Decision Letter 3]

13 Apr 2021

Infrared thermal imaging monitoring on hands when performing repetitive tasks: an experimental study

PONE-D-20-36722R3

Dear Dr. Garcia Alcaraz,

We’re pleased to inform you that your manuscript has been judged scientifically suitable for publication and will be formally accepted for publication once it meets all outstanding technical requirements.

Kind regards,

Benjamin A. Philip

Academic Editor

PLOS ONE